# Formation and Stability of Pea Proteins Nanoparticles Using Ethanol-Induced Desolvation

**DOI:** 10.3390/nano9070949

**Published:** 2019-06-29

**Authors:** Chi Diem Doan, Supratim Ghosh

**Affiliations:** Laboratory of Food Nanotechnology, Department of Food and Bioproduct Sciences, College of Agriculture and Bioresources, University of Saskatchewan, Saskatoon, SK S7N 5A8, Canada

**Keywords:** pea protein, alcohol-desolvation, nanoparticles, secondary structure, dispersion stability, accelerated gravitation

## Abstract

Protein nanoparticles have recently found a lot of interests due to their unique physicochemical properties and structure-functionality compared to the conventional proteins. The aim of this research was to synthesize pea protein nanoparticles (PPN) using ethanol-induced desolvation, to determine the changes in secondary structures and the particle stability in an aqueous dispersion. The nanoparticles were prepared by diluting 3.0 wt% pea protein solutions in 1–5 times ethanol at pH 3 and 10 at different temperatures. Higher ratios of ethanol caused greater extent of desolvation and larger sizes of PPN. After homogenization at 5000 psi for 5 min, PPN displayed uniform size distribution with a smaller size and higher zeta potential at pH 10 compared to pH 3. PPN prepared from a preliminary thermal treatment at 95 °C revealed a smaller size than those synthesized at 25 °C. Electron microscopy showed roughly spherical shape and extensively aggregated state of the nanoparticles. Addition of ethanol caused a reduction in β-sheets and an increase in α-helices and random coil structures of the proteins. When PPN were separated from ethanol and re-dispersed in deionized water (pH 7), they were stable over four weeks, although some solubilization of proteins leading to a loss in particle size was observed.

## 1. Introduction

Protein nanoparticles have gained a lot of interests in the food industry due to their small size and high surface to volume ratio. Joye and McClements [1] defined nanoparticles with at least one dimension smaller than 1000 nm, but larger than a single atom; however, the threshold of nanoparticle size depends on different materials and is not universally applicable. The extremely small size of the nanoparticles enables them to improve their dispersibility and stability to gravitational separation, aggregation, and applied stress during food processing. They also show excellent encapsulation ability by protecting a core bioactive from degradation and controlling its delivery and release at targeted sites within the human body [2]. Typically proteins used for the synthesis of nanoparticles are from animal origin, e.g., whey protein [3] or its component β-lactoglobulin [4].

In recent years, plant proteins (e.g., proteins extracted from soy, pulses, and cereals) have demonstrated advantages over the animal proteins (whey, casein, and gelatin) as an alternative “green” material in various food applications [5]. Plant proteins not only possess good amphiphilic and functional properties (emulsifying and foaming capacity), but they are also less allergenic compared to animal proteins [6,7]. Pea proteins (PP, isoelectric point (p*I*) ≈ 4.3), extracted from pea seeds, are one of the most popular plant proteins. They are mainly comprised of globulins (65–80%) and two minor fractions of albumins and glutelins [8]. Pea globulins (globular proteins) are mainly constituted of legumin (hexameric 11S globulin, with a molar mass ranging from 350 to 400 kDa, containing disulfide bridges), which is rich in β-sheet structure. Another minor fraction in pea globulins is a glycoprotein called vicilin (trimeric 7S globulin, with a molar mass of approximately 150 kDa) [9]. PP reveals a high solubility (≈ 80%) at pH values further away from their p*I* (e.g., at pH 2 and 9), whereas only 30% of PP are soluble at pH around 5 [10]. The capacity of PP in gelation [11], film formation [12], foaming [13], and emulsion stabilization [14] has been well-studied. Attempts have also been made to develop nano- or micro-particles from PP. For example, Irache et al. [15] successfully utilized a single fraction legumin of PP to create nanoparticles (mean particle diameter between 200 and 700 nm) with glutaraldehyde as a crosslinking agent. Whereas, when whole PP was used, microparticles obtained via coacervation varied from about 10 to hundreds of µm [16].

Besides these techniques, desolvation has also been proven as one of the promising methods for the synthesis of protein nanoparticles from β-lactoglobulin [17], whey protein [3], and bovine serum albumin (BSA) [18]. Desolvation is associated with the addition of an antisolvent agent (e.g. acetone, ethanol) into the protein solutions to induce supersaturation. Depending on solvent/antisolvent combinations, protein-protein interactions become strong enough to overcome protein-solvent interactions and entropy of mixing effects, leading to a decrease in solvation power and increase in supersaturation beyond a critical point, such that nucleation of protein particle is induced. As the nuclei formation continues, the dissolved protein concentration is decreased and below a critical supersaturation concentration, nucleation stops. However, the nuclei already formed continue to grow by condensation of protein molecules or aggregation of nuclei to form protein nanoparticles [1]. It is essential that a sufficiently strong repulsion remains between the protein particles to prevent them from excessive aggregation. Therefore, desolvation is usually conducted under basic pH to increase electrostatic repulsion between the protein particles [17]. In addition, several other factors, such as ethanol-water desolvation ratio, temperature, and ionic strength of the medium may significantly affect protein nanoparticle formation [3,19].

In the present work, nanoparticles were prepared from pea protein isolates (PPI) using different desolvation conditions (ethanol amount, pH, and temperature) followed by high-pressure homogenization. The pea protein nanoparticles (PPN) were characterized by size, charge, and microstructure. Furthermore, for the first time, changes in protein secondary structures in the PPN were analyzed using Fourier Transform Infrared Spectroscopy (FTIR). Subsequently, the re-dispersibility and storage stability of the PPN was also investigated.

## 2. Materials and Methods

### 2.1. Materials

Pea protein isolates (PPI) was kindly donated by Nutri-Pea Limited (Southport, MB, Canada). According to the supplier, PPI components include 3.5% moisture (16 h at 100 ± 5 °C), 83.0% protein (Dumas − N × 6.25), 7.0% lipid (AOCS 996.06), and 6.0% ash (AOAC 923.03). Deionized Milli-Q™ water (Millipore Corporation, Burlington, MA, USA) was used for the preparation of protein solutions. Anhydrous ethyl alcohol (100% ethanol) was purchased from Commercial Alcohols, Brampton, ON, Canada. All acids and bases were obtained from Thermo Fisher (Edmonton, AB, Canada). All the other chemicals were purchased from Sigma Aldrich (Mississauga, ON, Canada).

### 2.2. Preparation of Protein Solution

Pea protein isolates (5.0 wt%) was dissolved in deionized water and mixed with 0.02 wt% sodium azide to prevent any microbial growth. The protein dispersion was adjusted to pH 9 using 1.0 M NaOH while being stirred on a magnetic stirrer. The dispersion was kept stirring overnight at 25 °C to allow complete hydration of the proteins. The dispersion was subsequently centrifuged at 4000 rpm for 15 min to remove the insoluble fractions (consisting of insoluble proteins, lipid, polysaccharides and minerals), and the soluble protein in the supernatant was collected for further experiments.

### 2.3. Determination of Protein Concentration

The protein content of the supernatant (3.0 wt% protein, solubility 76.9%) was then determined using Lowry method with a modified biuret reagent and bovine serum albumin as the standard [20]. Specifically, the biuret reagent was freshly prepared and diluted eight times with 2.3% sodium bicarbonate to form reagent A. An aliquot of 4 mL reagent A was added to 1 mL of protein solution, held for 10 min at 25 °C, followed by the addition of 0.125 mL of 2 N Folin-Ciocalteu reagent (Thermo Fisher, Canada), and mixed using a vortex mixer. The absorbance of the mixture was recorded immediately at 660 nm, and the protein concentration was calculated from the absorbance using a calibration curve constructed with BSA.

### 2.4. Synthesis of Pea Protein Nanoparticles (PPN)

A volume of soluble protein solution from the supernatant (pH 9) (from Section 2.2) was continuously stirred at 400 rpm while various amounts of ethanol (1 to 5 times the protein volume, denoted by 1× to 5×) were added dropwise at a rate of 1 mL/min. Three desolvation temperatures were applied to investigate the influence of heating on the size distribution of the PPN synthesized with 5× desolvation. Besides particle synthesis at 25 °C and 65 °C, the protein solution was also heated to 95 °C for 15 min, cooled down to 65 °C before addition of ethanol. After desolvation, the protein-ethanol-water dispersions reached pH 10 ± 0.15, from which one portion was adjusted to pH 3 using 1.0 M HCl to study the effect of pH on particle size. These dispersions were subsequently passed through a high-pressure homogenizer (EmulsiFlex-C3, Avestin Inc., Ottawa, ON, Canada) (5000 psi/5min, 10000 psi/5min, and 15000 psi/5min) for particle size reduction.

### 2.5. Particle Size Distribution and Surface Charge

The surface-weighted average diameters (d_32_) and size distributions of the PPN were measured using a static laser diffraction particle size analyzer (Mastersizer 2000, Malvern Instruments, Montreal, QC, Canada), using deionized water as the dispersion medium (refractive index is 1.465). The surface charge or zeta potential (z, mV) of the PPN was determined after appropriate dilution in deionized water with corresponding pH adjustment using a Zetasizer Nano-ZS90 (Malvern Instruments, Westborough, MA, USA).

### 2.6. Secondary Structure Using Attenuated Total Internal Reflectance—Fourier Transform Infrared Spectroscopy (ATR-FTIR)

FTIR analysis was carried out on freeze-dried powder using a Renishaw inVia Reflex Raman microscope (Renishaw, Gloucestershire, UK) equipped with an Illuminant FTIR microscope accessory (Smith’s Detection, Danbury, Connecticut, US) accessory that allows the spectra to be acquired using a diamond attenuated total reflection (36X-ATR) objective. All data were collected in the spectral range from 4000 to 650 cm^−1^ at a spectral resolution of 4 cm^−1^. The background spectrum was obtained using reflectance from an empty germanium crystal surface, and sample spectra were collected as an average of 512 scans. Different protein secondary structures (α-helices, β-sheets, β-turns, and random coils) were determined by identifying the amide I component peak frequencies according to the Fourier self-deconvolution (FSD) algorithm [21], followed by multi-peak fitting with Gaussian function using Renishaw WiRE3.3 software (Renishaw, Gloucestershire, UK). The relative amounts of different secondary structural elements were quantified from the fitted multicomponent peak areas.

### 2.7. Particle Morphology

A scanning electron microscope (SEM—model SU8010, Hitachi High-Technologies Canada, Inc. Toronto, ON, Canada) was used to characterize the surface morphology of the PPN. Air-dried PPN were mounted on a glass slide with scotch tape, and sputter coated with chromium. The image acquisition was made at different magnifications using 3000 volt accelerating voltage and 3600 µm working distance.

### 2.8. Re-Dispersibility of Pea Protein Nanoparticles

After desolvation at pH 10, the ethanol-water dispersions of PPN were centrifuged at 4000 rpm for 15 min. The supernatant was discarded to collect the protein particles from the sediment, which were subsequently re-dispersed in deionized water (0.5 wt% particles) at different pH values under stirring (400 rpm) at 25 °C. The re-dispersed particles were homogenized at 5000 psi/5 min and analyzed for particle size, and zeta potential. The storage stability of the homogenized PPN re-dispersions was recorded for 4 weeks at 25 °C. The Z average diameter and zeta potential of PPN in the re-dispersion were measured using a Zetasizer Nano-ZS90 (Malvern Instruments, Westborough, MA, USA) after appropriate dilution in deionized water and corresponding pH adjustment.

### 2.9. Accelerated Gravitational Sedimentation of Pea Protein Nanoparticles

The PPN re-dispersions were subjected to accelerated gravitation in a photocentrifuge (LUMiSizer, LUM Americas, Boulder, CO, USA). An aliquot of 400 µL of dispersion was transferred into a rectangular polycarbonate vial (8 mm × 2 mm) and centrifuged at 1000 rpm at 25 °C, while the transmission of an 865 nm laser beam scanning through the cells over the total length of the sample was collected every 60 s for 16 h. Analysis of the transmission profiles and calculation of sedimentation velocity were done using the SEPView software v 4.1 (LUM GmbH, Berlin, Germany). 

### 2.10. Statistical Analysis

All data are expressed as means ± standard deviation of three experimental replications (with at least two analytical replications) and were analyzed using one-factor analysis of variance (ANOVA) using SPSS Statistics 20 (IBM Canada Ltd., Markham, ON, Canada). The equality of variances was verified using Levene’s test prior to the usage of Tukey’s test to compare the mean values at *p* < 0.05 significance level.

## 3. Results

### 3.1. Nanoparticle Synthesis and Characterization

PPN were synthesized using a constant volume of 3.0 wt% soluble pea protein solution at pH 9 (collected from the supernatant of the centrifugation of 5 wt% PPI dispersion). The particle size (d_32_) and surface charge of the initial protein solution was 0.50 ± 0.01 µm and −46.07 ± 0.59 mV at pH 9, respectively. The desolvation was initially done at 25 °C to investigate the influence of alcohol to aqueous phase ratios on the average size and surface charge of PPN. The average surface-weighted diameter (d_32_) of the particles ranged from 0.2 to 30 µm depending on pH and desolvation ratios (Figure 1). Their particle size distribution is also reported in Appendix A.

The size and surface charge of the protein-based particles are strongly dependent on the nature of the protein and the desolvation conditions (pH, ionic strength, temperature, desolvation ratio, and solute concentration) [13]. At pH 10, the non-desolvating protein solution (0×) had a similar size to that at pH 9 of the initial protein solution. There was a reduction in size when ethanol was added at the 1× ratio. Further increase in the ethanol volume led to an increase in the size of PPN. Similar behavior was observed by other researchers for different proteins [3,18]. Under high-pressure homogenization with 5000 psi/5 min at pH 10, the average size of the soluble protein (0×) and all PPN decreased (Figure 1A). The homogenized PPN prepared at 1×, 2×, 3×, and 4× desolvation ratios were ranged from 0.35 to 0.58 µm and not significantly different from each other (*p* > 0.05). Whereas homogenized particles at 5× desolvation ratio were significantly larger than the others (1.90 ± 0.07 µm) (*p* < 0.05) (Figure 1A). Similar behavior was reported by Sadeghi et al. [18], where the sizes of BSA particles also increased, reached a maximum when the desolvation ratios increased from zero to 4×. The authors proposed that once a nucleus of a small protein particle forms, they continue to grow in the presence of more desolvating solvent.

At pH 3, particle size of non-desolvating protein solution (0×) dramatically increased compared to that at pH 10. This could be associated to protein aggregation caused by the addition of acid (Figure 1B). After ethanol desolvation at pH 3, particle size remained similar except for the particles at 1× desolvation which showed a significant increase in particle size. After homogenization, the particles at pH 3 exhibited larger average size than those prepared at alkaline pH value, for example, 0.46 ± 0.01 µm at pH 10 compared to 2.66 ± 0.43 µm at pH 3 at 2× desolvation ratio. The particle distribution after homogenization showed a significant shift towards smaller sizes for both pH values (Appendix A). However, even, after homogenization, all PPN dispersions showed bi-modal particle size distribution, which could be due to particle re-aggregation.

The surface charge of all particles at pH 3 showed positive charge as the amino groups were protonated, while the ones prepared at pH 10 possessed negative charge resulting from the deprotonation of –COOH groups (Figure 1C). The charge on the particles decreased with the increase of desolvation ratio up to 3×, and thereafter no change was observed. Similar behavior of decrease in zeta potential of soy protein nanoparticles with increase in alcohol concentration was also reported by Teng, et al. [22]. Presence of ethanol decreased the dielectric constant of the solution, which hindered the ionization of charge groups on the protein leading to a lowering of zeta potential. No significant effect of homogenization on surface charge of the particles was observed except at 1× desolvation. Absolute charge of the particles fabricated at pH 10 was higher than those at pH 3 which could be responsible for higher repulsion and lower particle size at pH 10 compared to the lower pH. At an alkaline pH, the hexameric structures of pea globulins were loose, their chains unfolded, and the charges became more accessible [23]. At an acidic pH, below the isoelectric point (p*I*), the COO^−^ groups of proteins were protonated, making them uncharged (COOH), and their zeta potential decreased [24].

The effect of high-pressure homogenization has been proven to reduce the particle size (Figure 1A,B) due to the aggregate disruption and disulfide-bond breaking [25]. As discussed above, the particles fabricated at 5× revealed the largest size even after homogenization. Hence, higher pressures (10000 psi/5 min, and 15000 psi/5 min) were exploited to test if the particle size could be further decreased (Figure 2A). However, higher homogenization pressure did not significantly reduce (*p* > 0.05) the particle size at both pH values. It should be noted that the purpose of homogenization was not to break the individual particles into smaller ones, but to disrupt the particle aggregates. It could be possible that the disrupted particle aggregates were re-aggregating due to strong hydrophobic interactions before particle size analysis. Surface charge of the particles, on the other hand, showed a large increase at pH 10 from −28.63 ± 3.02 mV at 5000 psi to −56.55 ± 3.61 mV at 10,000 psi (Figure 2C). This might be associated with the exposure of more charged group on the PPN when a higher homogenization pressure enables unfolding of the hexameric pea globulin structures. At pH 3, the surface charge of the PPN showed a slight increase with homogenization pressure, but not significant (*p* > 0.05).

Figure 2B illustrates the average sizes of homogenized (5000 psi/5 min) 5× desolvated PPN prepared at different desolvation temperatures. The particles synthesized at 65 °C exhibited a decrease in size (1.02 ± 0.35 µm at pH 10) compared to the ones prepared at 25 °C (1.88 ± 0.47 µm at pH 10). Pre-heating the protein solutions at 95 °C and cooing to 65 °C before desolvation led to the formation of even smaller particles at both pH values (0.66 ± 0.15 µm at pH 10). The PPN formed at higher temperatures also revealed an increase in zeta potential at pH 10 compared to pH 3 (Figure 2D). The influence of temperature on desolvation process can be expressed in terms of rate of nucleation, and aggregation due to a change in the diffusion rate, viscosity of the medium, the equilibrium saturation and supersaturation concentration of the protein [1,26]. Heating also causes denaturation of proteins [27], resulting from the unfolding and stretching of molecular structures which exposes the interior hydrophobic groups leading to more accessibility of the charges (shown by increase in zeta potential) and a better solubilization in an antisolvent medium. Hence, particle formation and growth will occur at a slower rate resulting in the creation of smaller aggregates [28]. Previously, Gülseren et al. [3] reported a significant increase in particle size upon heating a whey protein (WP) suspensions up to 90 °C in the presence of ethanol due to hydrophobic interactions and disulfide bond formation. However, the researchers did not apply homogenization after particle formation, which could break apart loosely bound aggregates. In the present case, application of high-pressure homogenization after heating and desolvation allowed better control of particle size distribution and the formation of protein nanoparticles.

### 3.2. Particle Microstructure

Figure 3 illustrates the morphology of air-dried native pea protein solutions and 2× and 3× desolvated, homogenized (5000 psi/5 min) PPN at pH 3 and pH 10 (25 °C). Application of air-drying led to the appearance of aggregated particles under SEM. From Figure 3A,D, no particulate structure was observed in the native protein dispersion at both pH 3 and 10, indicating the nature of dry protein film from a solution. In contrast, PPN appeared to be roughly spherical, uneven and aggregated (Figure 3B,C,E,F). At pH 3, the PPN appeared to be larger than the particles formed at pH 10, which is consistent with the average particle size data reported in Figure 1. However, the size of individual particles from the SEM images appeared much smaller than that reported in average particle size data, which could be attributed to extensive particle aggregation hindering measurement of individual particle size using laser diffraction technique. Instead, the laser diffraction particle size analyzer determined the aggregate size observed in the SEM images. It has been reported that alcohol led to partial unfolding of proteins and opening of hydrophobic regions, leading to protein as well as protein particle aggregation [29]. Similar aggregated nanoparticle shape was also observed when ethanol was used as a desolvating agent for wheat gliadin [30].

### 3.3. Conformational Changes in the Secondary Structures of Pea Protein Nanoparticles

A full representative FTIR spectra of 3× desolvated pea protein particles at 25 °C is illustrated in Appendix A. The modifications in pH, temperatures, desolvation ratios, and homogenization pressures during processing led to conformational changes, indicated by the difference in intensities of the amide I band peaks. Secondary structure components of raw PPI powder, and the supernatant of PPS at pH 10 and pH 3 without any alcohol desolvation are shown in Appendix A. Raw PPI contains a high percentage of β-sheets (54%), a similar quantity of α-helices (24%) and β-turns (20%), and a small amount of random coils (RC 2%), indicative of compact globular protein structure. After removal of more hydrophobic insoluble components, the average secondary structure of the supernatant (PPS) at pH 10 showed an increase in β-sheets and α-helices, while β-turns decreased and no RC was found. Acidification of the supernatant PPS (pH 3) caused a significant increase in α-helices at the expense of β-sheets and β-turns structures, which could be due to the neutralization of acidic amino acids by protonation, resulting in a decrease in negative charge and formation of more local hydrogen bonded α-helical structures [31].

Figure 4A,B illustrates the influence of ethanol on the secondary structure contents in PPN at pH 10 and pH 3, respectively. There was a significant increase in α-helices and RC at the expense of a significant reduction in β-sheets when the desolvation ratios increased from 0× to 3×. However, a further surge in ethanol amounts (4× and 5×) at pH 10 led to a slight decrease in α-helices, and no change in β-sheets and RC. Ethanol is also a lower dielectric constant solvent, stronger hydrogen bond donor, but weaker acceptor than water. Therefore, in the presence of ethanol, the protein carbonyl group will become stronger acceptor and their ability to form local intramolecular hydrogen bond with amide donors will enhance. Such intramolecular hydrogen bond will increasingly favor the formation of α-helical structure at the expense of β-sheets [32]. Early findings on the promotion of α-helices by alcohols have also been observed in human myoglobin [33], and β-casein [34]. Alcohols are also known to weaken non-local hydrophobic interactions that hold β-sheet structures. However, once a large amount of α-helices is formed, further addition of alcohol may not lead to the generation of α-helical structure, hence a plateau-like behavior in α-helices was observed at 4× and 5× desolvation ratios. The presence of alcohol may also lead to protein denaturation and an increase in random coil as observed in Figure 4A. 

At pH 3, a significant drop in α-helices and increase in β-sheets was observed for 1× compared to the sample without any ethanol (0×) (Figure 4B). This behavior contrasts with the commonly observed preference towards α-helices in the presence of alcohol. This could be due to the way PPN were made. The non-desolvated pea proteins (0×) were directly acidified from pH 9, while the desolvated PPN were initially formed at pH 10 in the presence of ethanol and then acidified to reach pH 3 (they were not desolvated at pH 3). Changing to pH 3 after the desolvated particles were formed did not significantly influence their α-helices compared to pH 10, while the β-sheets increased at the expense of β-turns (Figure 4B).

At pH 3, with increase in desolvation ratio, a gradual increase in α-helical structure was formed at the expense of β-sheet structure. At a higher desolvation ratios (4×–5×), formation of α-helices of PPN was stronger at acidic conditions compared to the alkaline conditions; e.g. for 5× desolvation, 48.8% α-helices at pH 3 versus 35.6% at pH 10. This finding is in agreement with the report of Tsumura et al. [31], who investigated the changes in secondary structures of soybean β-conglycinin using far-UV circular dichroism. The quantity of β-turn also showed an increase with ethanol desolvation ratio at pH 3. Reverse turns in proteins represent sites of chain reversal, therefore, an increase in β-turn signifies an increase in the globular characteristics of the protein in the presence of acidic ethanol [35]. Unlike pH 10, increase in acidic ethanol also did not show any significant increase in RC structure until 5× ethanol desolvation ratio, which indicates less protein denaturation under acidic pH compared to basic pH. Proteins with increased RC structures have been shown to have better functional properties such as emulsification and thickening as their structure allows them to bind to larger amounts of water molecules or other chemical moieties [36]. Therefore, it is expected that the PPN prepared at pH 10 would be better emulsifier compared to PPN prepared at pH 3.

The influence of temperature on the secondary structure of 5X ethanol-induced pea proteins was also analyzed (Figure 4C,D). At basic pH, an increase in desolvation temperature led to a reduction in α-helices and RC, and a surge in β-sheets and β-turns. A similar result was reported by Selling et al. [37], who confirmed a loss in α-helices upon heating zein from 25 to 70 °C in a 80% methanol solution. There was no explanation why heating changed the secondary structure of zein; however, the authors also noticed changes in tertiary structure of zein upon heating resulting in a change in the local environment of the aromatic amino acid residues. In the current research, the influence of heating at acidic pH, however, increased the amount of α-helices, β-turns and RC with the expense of decrease in β-sheets. Clearly, heating protein during or before desolvation can alter the secondary structures of protein where the extents of the changes depend on temperature, protein type, pH, and desolvation ratio.

### 3.4. Storage Stability of Pea Protein Particles

To use the PPN in a variety of food and beverage application, it is important to have a stable dispersion without sedimentation or dissolution. To investigate the storage stability of the PPN, they were separated from ethanol, re-dispersed in deionized water at different pH values and homogenized at 5000 psi for 5 min. Only the particle synthesized at pH 10 was used for this study as they were smaller and more stable compared to the particle synthesized at pH 3 (Figure 2). Various pH values were used so that a suitable pH could be identified where the nanoparticles were the most stable against sedimentation. Re-dispersibility of 2× PPN is shown in Figure 5A as a function of pH ranging from 2 to 10. At low pH values (pH 2–5), the particles aggregated and settled at the bottom of the glass vials, whereas the rest of dispersions at pH values 6–10 displayed homogeneous behavior. Dispersion stability at higher pH values can be attributed to strong electrostatic repulsions between the PPN, indicated by the significantly higher zeta potential of the re-dispersed nanoparticles from pH 6 to 10 (Figure 5B). 

Quantitative estimation of dispersion stability at various pH values was obtained by calculating sedimentation velocity from the photocentrifuge transmission profiles (Figure 5C). The visually stable dispersions (pH 6–10) were separated gradually upon centrifugation, while the unstable dispersions (pH 2–5) were separated almost instantaneously. Hence sedimentation velocity was calculated only for the sample with pH 6 to 10. The dispersion at pH 7 was the most stable with least sedimentation velocity (50.5 µm/min at 1000 rpm), followed by the dispersion at pH 6. Interestingly, the dispersions at pH 8–10 showed higher sedimentation velocities (102–105 µm/min), although their zeta potentials were higher than the others. This shows that the surface charge is not the only factor responsible for PPN dispersion stability. Other factors, such as particle size, density, and protein conformation could also be important.

From the above discussion, it is apparent that the most stable re-dispersion of PPN were obtained at pH 7. Therefore, long-term stability for all PPN re-dispersions (prepared with different desolvation ratios) was recorded at pH 7 during 4-week storage at 25 °C (Figure 6A). The particle size of the re-dispersed PPN was too small to analyze using a static laser diffraction analyzer. Hence a dynamic light scattering device was used. The decrease in particle size after re-dispersion could be attributed to the change in the solvent from ethanol-rich to a simple aqueous phase which could solubilize a part of the nanoparticles. Also, re-dispersion in aqueous phase led to loss of aggregated particle structure and it was possible to measure the actual nanoparticle size. There was no significant difference in the Z-average particle size between the freshly re-dispersed 2×, 3×, 4×, and 5× PPN, which were between 333–337 nm, higher than the size of re-dispersed 1× PPN (267 nm) (Figure 6A). Probably the 1× desolvated particles were more soluble leading to a significant loss of particle morphology. After 4-week storage at 25 °C, there was a reduction on the average particle sizes, which might also be due to the dissolution of larger PPN. However, this decline was not statistically significant (*p* > 0.05) except for 1× and 5× PPN. Quantitative estimation of stability of all the PPN re-dispersions at pH 7.0 was obtained by measuring the rate of sedimentation under accelerated gravitation (1000 rpm, ~130 g) (Figure 6B). As expected, lowest sedimentation rate was obtained for 1× desolvated PPN re-dispersion as their particle size was lowest, and the rate increased with an increase in desolvation ratio. Microstructure of the re-dispersed PPN at pH 7 also displayed the appearance of aggregated nanoparticle structure (Appendix A), which confirms that at pH 7 the PPN structure remained stable and they were not completely soluble.

A high ability to be re-dispersed in the aqueous phase with pH >6 provides a prospect for the PPN to stabilize Pickering emulsions as well as a matrix for bioactive delivery. Research on such applications of PPN at pH 7 is currently underway. However, for similar application of PPN at acidic pH, further research is needed to improve their stability. 

## 4. Conclusions

Nanoparticles from globular pea protein isolates were successfully synthesized for the first-time using ethanol desolvation followed by high-pressure homogenization. The sizes of PPN can be tailored by changing the ethanol to water ratio, pH of the medium, and pre-heating the dispersion before desolvation. The PPN showed smaller sizes and higher surface charges at pH 10 than at pH 3. Nanoparticles synthesized from pre-heated protein solutions showed higher surface charge compared to the nanoparticle fabricated at 25 °C. The desolvation led to a roughly spherical shape and extensive aggregation of PPN, as well as a decrease in β-sheet content and an increase of α-helix and random coil structures of the protein. The PPN, recovered from ethanol solution and re-dispersed in aqueous phase, showed better stability at pH > 6 (best at pH 7), but a decrease in size indicated loss of protein due to dissolution. The re-dispersed PPN at pH 7 remained stable during a 4-week study with a minor decrease in particle size. The PPN could be used to stabilize Pickering oil-in-water emulsions and as a matrix for bioactive delivery.

## Figures and Tables

**Figure 1 nanomaterials-09-00949-f001:**
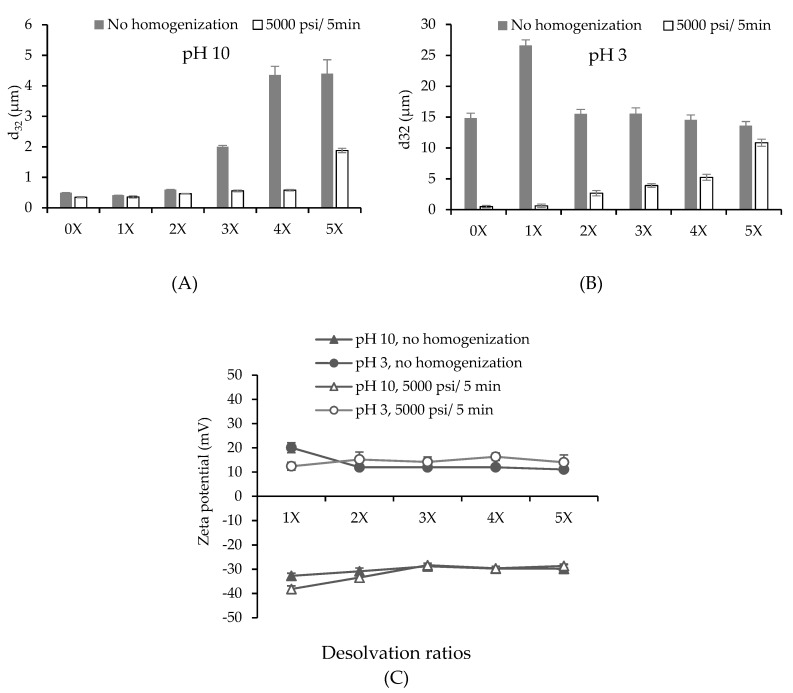
(**A**,**B**) Effect of ethanol amount on surface average droplet diameter (d_32_) at pH 10 (**A**) and pH 3 (**B**) before and after homogenization at 5000 psi for 5 min. (**C**) Surface charge of pea protein particles as a function of desolvation ratios at pH 10 and pH 3 before and after high-pressure homogenization.

**Figure 2 nanomaterials-09-00949-f002:**
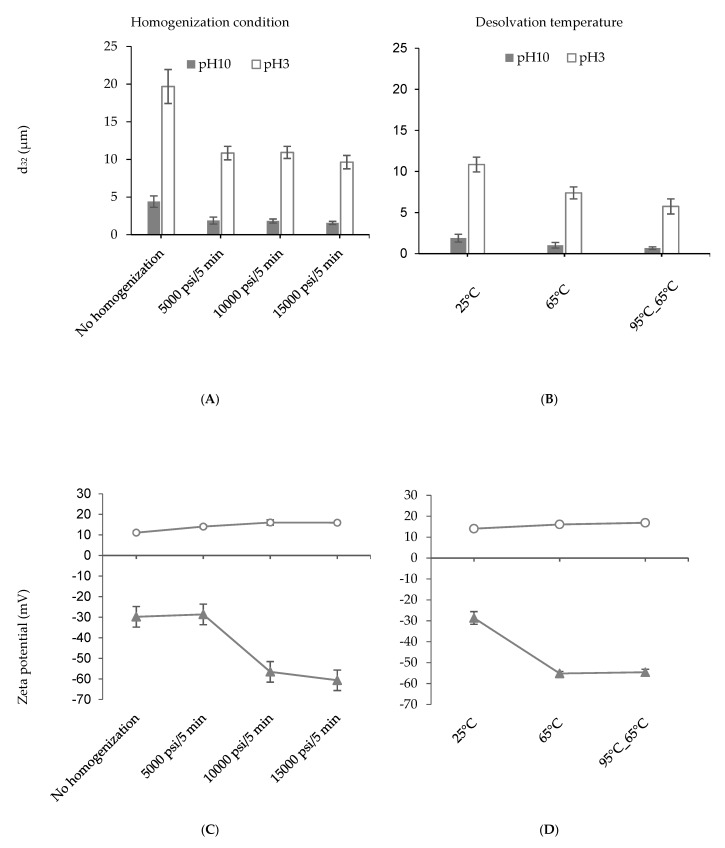
Influence of homogenization pressure (**A**,**C**) and heating (**B**,**D**) on average particle size (**A**,**B**), and surface charge (**C**,**D**) of pea protein particles at pH 10 (close triangles) and pH 3 (open circles). Pea protein particles were prepared with 5× desolvation ratio of alcohol to protein solution.

**Figure 3 nanomaterials-09-00949-f003:**
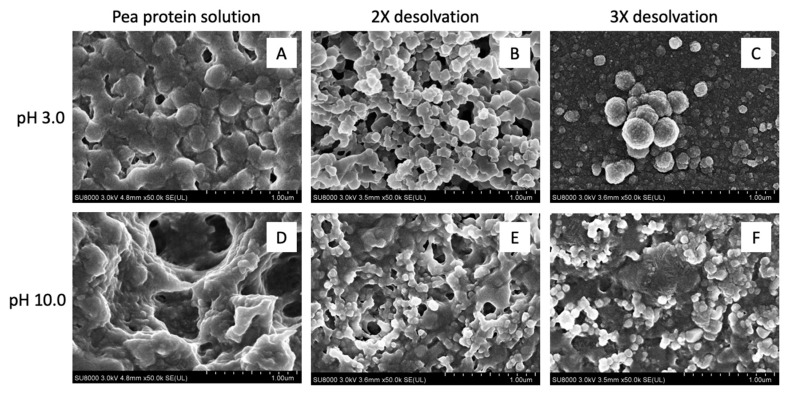
Microstructures of pea protein solutions (**A**,**D**) and 2× (**B**,**E**) and 3× (**C**,**F**) ethanol-desolvated nanoparticles at pH 3 (**A**–**C**) and pH 10 (**D**–**F**) obtained using a scanning electron microscope under 50,000× magnification. The protein particles were prepared at 25 °C and homogenized at 5000 psi for 5 min before air drying on the glass slide.

**Figure 4 nanomaterials-09-00949-f004:**
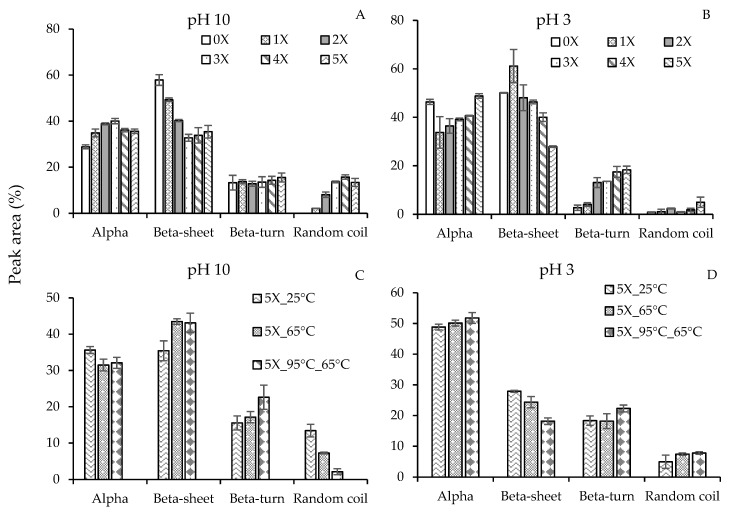
Estimation of secondary structure contents (%) of ethanol-induced pea proteins nanoparticles at (**A**,**C**) pH 10 and (**B**,**D**) pH 3 at 25 °C by FTIR-analysis as a function of different (**A**,**B**) desolvation ratios (ethanol: protein) and (**C**,**D**) desolvation temperatures.

**Figure 5 nanomaterials-09-00949-f005:**
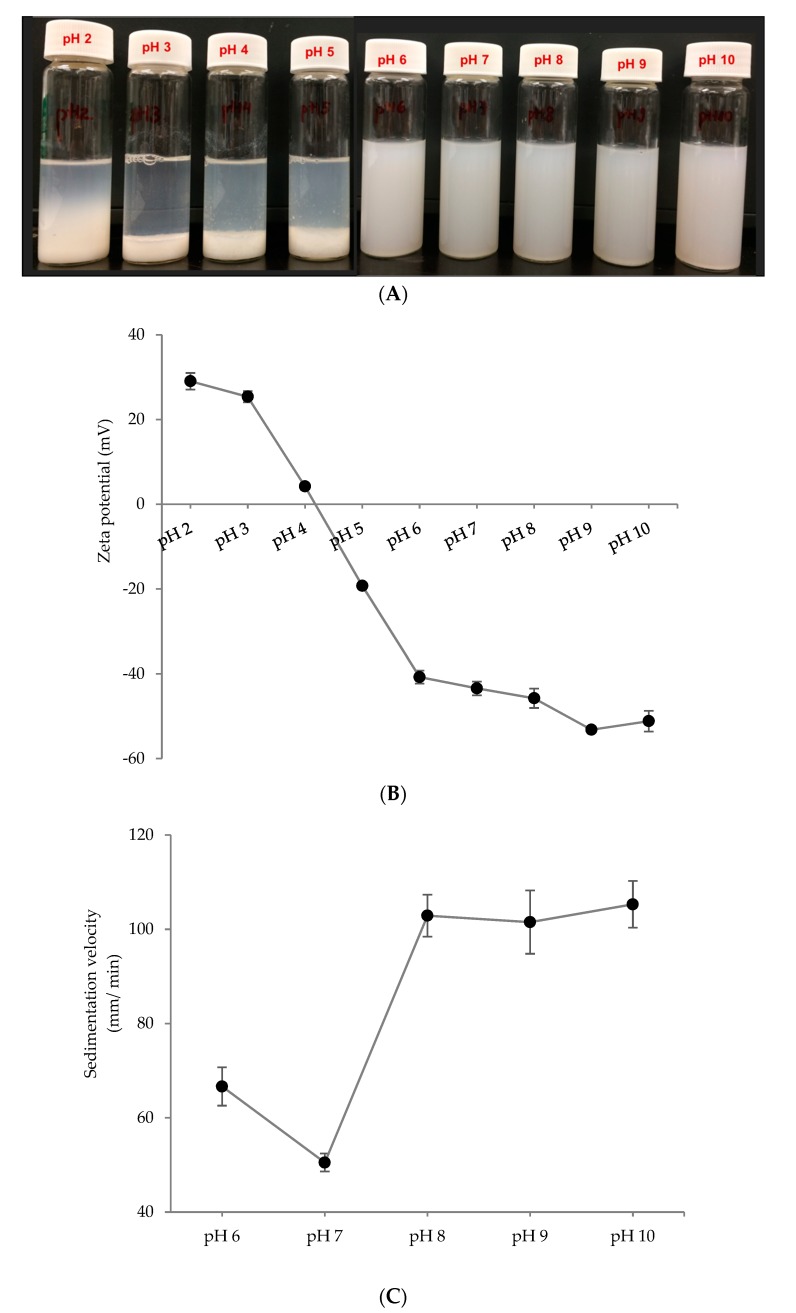
Stability of re-dispersed 2×-desolvated pea protein nanoparticles at different pH values. (**A**) visual observation, (**B**) zeta potential, and (**C**) sedimentation velocity (µm/ min) measured at 1000 rpm (~130 g) in a photocentrifuge.

**Figure 6 nanomaterials-09-00949-f006:**
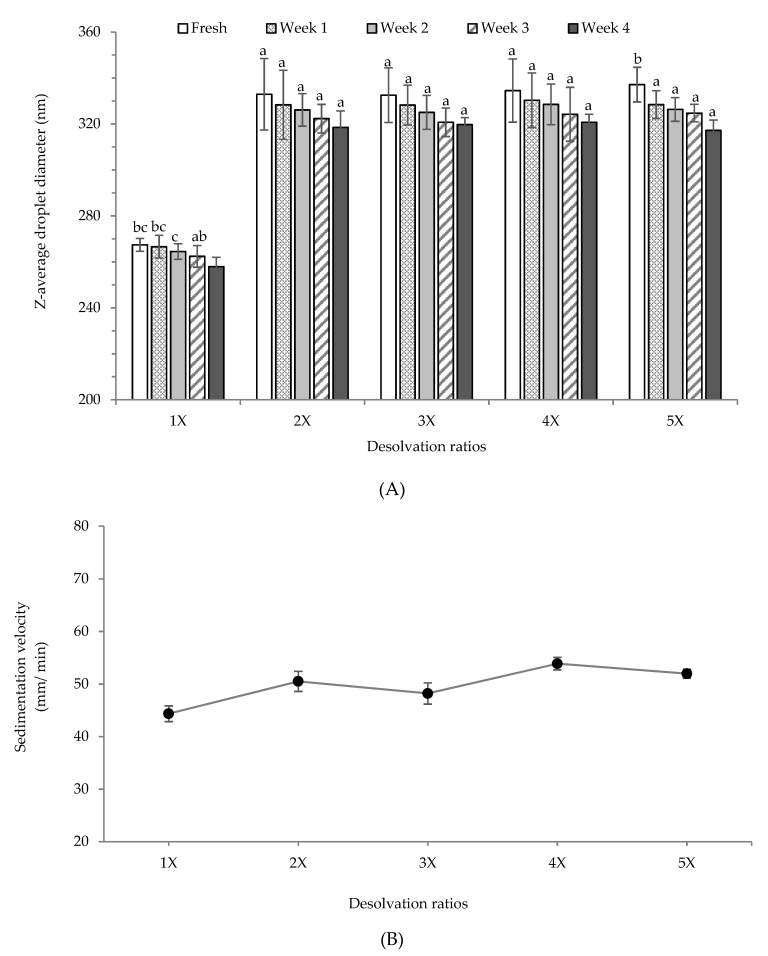
(**A**) Storage stability of re-dispersed pea protein nanoparticles prepared with different ethanol desolvation ratios at pH 7, (**B**) sedimentation velocity (µm/ min) of the freshly re-dispersed pea protein nanoparticles at pH 7 under accelerated gravitation (1000 rpm, ~130 g).

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
