# Peer review of "Formation and Stability of Pea Proteins Nanoparticles Using Ethanol-Induced Desolvation"

_nanomaterials, 2019, doi:10.3390/nano9070949_

Reviewer 1 Report

This manuscript reports aggregated protein particles which can be prepared by ethanol-induced desolvation of pea proteins. The fundamental aspects of the particles including size, zeta-potential and long-term stability are comprehensively shown. The manuscript is suitable for publication in Nanomaterials pending several points below.

1. The crude pea protein isolates contain impurities including lipids. The authors should discuss more clearly the possible effect of lipids on the particle formation.

2. In Figure 2, C and D (zeta-potential), the explanations of keys (filled triangles and empty circles) should be explained. The reason for the decrease in the zeta-potential values at high pressures (Figure 2C) and at high temperatures (Figure 2D) is nuclear. Please clarify.

3. The section 3.3 is a bit lengthy. The sentence “When the HCl was added to… the same without any ethanol” (lines 312-314 on page 9) seems not to be consistent with the results shown in Figure 4.

4. Can the authors’ approach be applied to the preparation of particles of biologically active proteins (enzymes)? Possible applications of PPN are recommended to be discussed in more detail.

Author Response

This manuscript reports aggregated protein particles which can be prepared by ethanol-induced desolvation of pea proteins. The fundamental aspects of the particles including size, zeta-potential and long-term stability are comprehensively shown. The manuscript is suitable for publication in Nanomaterials pending several points below.

1. The crude pea protein isolates contain impurities including lipids. The authors should discuss more clearly the possible effect of lipids on the particle formation.

Response: The particle formation was synthesized using the soluble pea protein solution after discarding the insoluble fraction via centrifugation, in which the lipid fraction was present. The lipid cannot be solubilized in water. Hence, there is no lipid fraction in the soluble pea protein solution, and hence we believe that there is no possible effect of lipids on particle formation. This is now mentioned in line 92-93.

2. In Figure 2, C and D (zeta-potential), the explanations of keys (filled triangles and empty circles) should be explained. The reason for the decrease in the zeta-potential values at high pressures (Figure 2C) and at high temperatures (Figure 2D) is nuclear. Please clarify.

Response: The explanation of the keys in Figure 2 was added. An attempt to explain the increase in zeta potential of PPN due to high pressures (line 243-244) and high temperatures was added (line 277-278).

3. The section 3.3 is a bit lengthy. The sentence “When the HCl was added to… the same without any ethanol” (lines 312-314 on page 9) seems not to be consistent with the results shown in Figure 4.

Response: The section 3.3 has been edited to make it shorter and easy to read. The sentence mentioned by the reviewer was modified to make it clear so that to would be easy to match with the results in Figure 4B. This is now line 344-345.

4. Can the authors’ approach be applied to the preparation of particles of biologically active proteins (enzymes)? Possible applications of PPN are recommended to be discussed in more detail.

Response: In theory, this approach can be applied to any protein being soluble in water. However, it is important to insist that the addition of ethanol can change the secondary structures of the protein, and maybe the activity of the enzymes. Hence, it is not recommended to use alcohol denaturation to create enzyme nanoparticles.

In the conclusion, possible applications of PPN was mentioned in the last statement. In practical, we have finished the research relating to the application of PPN, which will be published soon. We hope to show interesting and useful applications of PPN in our future publications.

Reviewer 2 Report

Doan and Ghosh describe the preparation of pea proteins nanoparticles and a deep the study of their properties  Experimental work is well conducted and results are clearly described. Thus, in my opinion the article can be considered for its publication in Nanomaterials after minor revision. Some mistakes/suggestions are detailed below:

Page 1, line 33, please replace showed by show

Page 5, line 189, please write Sadeghi et al.

Page 5, lines 194-197. Do the authors find an explanation for this result?

Page 7, line 247, please write Gülseren et al.

Page 8, line 279, please delete "the" (processing led to conformational changes)

Page 8, line 295, please write lower instead low

Page 9, lines 312-320. It results confusing the change in he preparation of the PPN. It is not clear why the preparation methodology was changed,

Page 10, line 338, please write Tsumura et al

Author Response

Doan and Ghosh describe the preparation of pea proteins nanoparticles and a deep the study of their properties.  Experimental work is well conducted and results are clearly described. Thus, in my opinion the article can be considered for its publication in Nanomaterials after minor revision. Some mistakes/suggestions are detailed below:

Page 1, line 33, please replace showed by show

Response: corrected, now line 33.

Page 5, line 189, please write Sadeghi et al.

Response: corrected, now line 206.

Page 5, lines 194-197. Do the authors find an explanation for this result?

Response: We have edited the lines to highlight acid-induced particle aggregation (line 211), however we offer no explanation of sudden increase in particle size at 1X desolvation.

Page 7, line 247, please write Gülseren et al.

Response: corrected, now line 280.

Page 8, line 279, please delete "the" (processing led to conformational changes)

Response: corrected, now line 312.

Page 8, line 295, please write lower instead low

Response: corrected, now line 327.

Page 9, lines 312-320. It results confusing the change in the preparation of the PPN. It is not clear why the preparation methodology was changed,

Response: The preparation methodology was not changed. We followed the same methodology as describe in the materials and methods section. Our writing was not clear. We now have edited that line and made it simpler: “At pH 3, a significant drop in α-helices and increase in β-sheets was observed for 1X compared to the sample without any ethanol (0X) (Figure 4B).” Now line 344-345.

Page 10, line 338, please write Tsumura et al

Response: corrected, line 356.